# A synthetic peptide that prevents cAMP regulation in mammalian hyperpolarization-activated cyclic nucleotide-gated (HCN) channels

Andrea Saponaro[1], Francesca Cantini[2,3], Alessandro Porro[1], Annalisa Bucchi[1], Dario DiFrancesco[1], Vincenzo Maione[4], Chiara Donadoni[1], Bianca Introini[1], Pietro Mesirca[5,6], Matteo E Mangoni[5,6], Gerhard Thiel[7], Lucia Banci[2,3,4,8], Bina Santoro[9], Anna Moroni[1,10]*

[1]Department of Biosciences, University of Milan, Milan, Italy; [2]Department of Chemistry, University of Florence, Florence, Italy; [3]Magnetic Resonance Center, University of Florence, Florence, Italy; [4]Interuniversity Consortium for Magnetic Resonance of Metalloproteins, Sesto Fiorentino, Italy; [5]Institut de Génomique Fonctionnelle, CNRS, INSERM F-34094, Université de Montpellier, Montpellier, France; [6]Laboratory of Excellence Ion Channels Science and Therapeutics, Valbonne, France; [7]Department of Biology, TU-Darmstadt, Darmstadt, Germany; [8]Institute of Neurosciences, Consiglio Nazionale delle Ricerche, Florence, Italy; [9]Department of Neuroscience, Columbia University, New York, United States; [10]Institute of Biophysics, Consiglio Nazionale delle Ricerche, Milan, Italy

*For correspondence:
anna.moroni@unimi.it

**Competing interests:** The authors declare that no competing interests exist.

**Abstract** Binding of TRIP8b to the cyclic nucleotide binding domain (CNBD) of mammalian hyperpolarization-activated cyclic nucleotide-gated (HCN) channels prevents their regulation by cAMP. Since TRIP8b is expressed exclusively in the brain, we envisage that it can be used for orthogonal control of HCN channels beyond the central nervous system. To this end, we have identified by rational design a 40-aa long peptide (TRIP8b$_{nano}$) that recapitulates affinity and gating effects of TRIP8b in HCN isoforms (hHCN1, mHCN2, rbHCN4) and in the cardiac current $I_f$ in rabbit and mouse sinoatrial node cardiomyocytes. Guided by an NMR-derived structural model that identifies the key molecular interactions between TRIP8b$_{nano}$ and the HCN CNBD, we further designed a cell-penetrating peptide (TAT-TRIP8b$_{nano}$) which successfully prevented β-adrenergic activation of mouse $I_f$ leaving the stimulation of the L-type calcium current ($I_{CaL}$) unaffected. TRIP8b$_{nano}$ represents a novel approach to selectively control HCN activation, which yields the promise of a more targeted pharmacology compared to pore blockers.

DOI: https://doi.org/10.7554/eLife.35753.001

## Introduction

Hyperpolarization-activated cyclic nucleotide-gated (HCN1-4) channels are the molecular correlate of the $I_f$/$I_h$ current, which plays a key role in controlling several higher order physiological functions, including dendritic integration and intrinsic rhythmicity both in cardiac and neuronal cells (*Robinson and Siegelbaum, 2003*). Unique among the voltage-gated ion channel superfamily, HCN channels are modulated by the direct binding of cAMP to their cyclic nucleotide binding domain (CNBD). Binding of the cyclic nucleotide increases the channel open probability upon

hyperpolarization via conformational changes in the CNBD that are propagated to the pore through the C-linker domain (*DiFrancesco and Tortora, 1991*; *Wainger et al., 2001*; *Zagotta et al., 2003*).

In addition to cAMP, HCN channels are regulated by TRIP8b, a brain-specific auxiliary (β) subunit, which modulates two independent features of the channel, namely trafficking and gating (*Santoro et al., 2009*; *Zolles et al., 2009*). For this dual regulation, TRIP8b binds HCN channels through two distinct sites: via the tetratricopeptide repeat (TPR) domain, which interacts with the last three amino acids (SNL) of HCN channels and regulates their trafficking; and via the TRIP8b$_{core}$ domain, which interacts with the CNBD and antagonizes the effect of cAMP on the voltage dependency of the channel (*Santoro et al., 2011*; *Han et al., 2011*; *Hu et al., 2013*).

Here, we focus our attention on the specific action of TRIP8b in preventing cAMP regulation of HCN channels. Given the brain-specific localization of TRIP8b, we posit that a TRIP8b-derived peptide drug, able to reproduce the effect of the full length protein on HCN channel gating, can be developed for orthogonal selective regulation of HCN in cells/tissues in which TRIP8b is not expressed. cAMP-dependent modulation of HCN channels underlies distinct roles of cAMP in heart rate regulation (*DiFrancesco, 1993*) and development of peripheral neuropathic pain (*Emery et al., 2012*; *Herrmann et al., 2017*), which can be dissected by using a TRIP8b-based tool. In this regard, peptide-based drugs (2–50 aa long) are emerging as a fascinating application area as they open new therapeutic possibilities with an advantage over small molecules in terms of specificity and affinity for the target (*Fosgerau and Hoffmann, 2015*; *Henninot et al., 2018*). To this end, we searched for the minimal peptide that binds to the CNBD and recapitulates the gating effect of full length TRIP8b in three HCN isoforms (HCN1, HCN2 and HCN4) and in the native I$_f$ current. In previous studies, we identified the core portion of TRIP8b (TRIP8b$_{core}$, 80 aa long) that interacts with the HCN CNBD and prevents cAMP modulation in full length channels (*Santoro et al., 2011*; *Hu et al., 2013*; *Saponaro et al., 2014*). A recent paper (*Lyman et al., 2017*) reported an even shorter binding sequence of TRIP8b (37 aa). However, this peptide, which was identified by progressive truncation of TRIP8b$_{core}$, failed to reproduce the binding affinity of the starting construct. Moreover, evidence for activity of this peptide on HCN currents is lacking. In the present study, we adopted a structure-driven rational design approach to engineer a 40-aa long peptide, TRIP8b$_{nano}$, that efficiently prevents cAMP regulation of HCN channels. The rational design of this peptide, based on secondary structure predictions and on NMR data of TRIP8b$_{core}$, was supported by an NMR-based 3D model structure of the complex formed by the TRIP8b$_{nano}$ peptide and CNBD of the human HCN2 channel isoform. This structural information identifies crucial interactions between the two partners and explains both direct (*Han et al., 2011*; *DeBerg et al., 2015*; *Bankston et al., 2017*) and indirect (allosteric) (*Hu et al., 2013*; *Saponaro et al., 2014*) modes of competition between TRIP8b and cAMP for binding to the CNBD. The evidence that TRIP8b$_{nano}$ establishes all relevant interactions with the CNBD is reflected by the finding that, contrary to shorter core sequences (*Lyman et al., 2017*), it binds to the isolated CNBD with the same affinity as TRIP8b$_{core}$ and acts with even higher efficacy than TRIP8b$_{core}$ in preventing cAMP modulation of full length HCN channels (*Hu et al., 2013*). In pacemaker myocytes of the sino-atrial node (SAN), TRIP8b$_{nano}$ equally prevented cAMP stimulation of native f-channels leading to a 30% reduction in spontaneus firing rate.

To develop TRIP8b$_{nano}$ as a membrane permeable drug, we linked it with the positively charged TAT sequence (*Herce et al., 2014*). TAT-TRIP8b$_{nano}$ was tested in SAN pacemaker myocytes where its addition to the extracellular buffer prevented adrenergic stimulation of the I$_f$ current leaving the activation of the L-type calcium current (I$_{CaL}$) unaffected. Our study opens the possibility of selective in vivo control of the cAMP-dependent facilitation of HCN channel opening, by local supply of TAT-TRIP8b$_{nano}$ peptide.

## Results

We have previously shown that TRIP8b$_{core}$ (residues 223–303 of mouse TRIP8b splice variant 1a4, hereafter TRIP8b) interacts with two elements of the isolated CNBD protein fragment from HCN channels (residues 521–672 of human HCN2, hereafter CNBD): the C-helix and the N-bundle loop, a sequence connecting helix E' of the C-linker with helix A of the CNBD (*Saponaro et al., 2014*). Biochemical assays confirmed that each of these two elements, that is, the N-bundle loop and C-helix, is necessary but not sufficient for binding (*Saponaro et al., 2014*).

To understand the interaction in atomic detail, we used solution NMR spectroscopy to characterize the structural properties of the CNBD - TRIP8b$_{core}$ complex. However, the NMR spectra of TRIP8b$_{core}$ showed very few signals. In order to improve the quality of the NMR spectra, we reduced the length of the TRIP8b fragment by progressively removing residues at the N- and C-termini with no predicted secondary structure. The truncated peptides were then tested for CNBD binding activity by isothermal titration calorimetry (ITC). We thus identified a 40-aa peptide (TRIP8b$_{nano}$, comprising residues 235–275 of TRIP8b, *Figure 1A*) with a binding K$_D$ of 1.5 ± 0.1 µM, a value similar to the K$_D$ of 1.2 ± 0.1 µM obtained with TRIP8b$_{core}$ (*Figure 1B*). TRIP8b$_{nano}$ was therefore employed for all subsequent NMR experiments, resulting in a remarkable improvement in the spectral quality and sample stability.

## Structural characterization of TRIP8b$_{nano}$ bound to CNBD

The comparison of the $^1$H-$^{15}$N HSQC spectra of TRIP8b$_{nano}$ with and without CNBD bound shows that the peptide folds upon interaction with the CNBD. Thus, the $^1$H-$^{15}$N HSQC spectrum of TRIP8b$_{nano}$ without CNBD shows a limited $^1$H resonance dispersion, characteristic of unstructured proteins (*Dyson and Wright, 2004*), while a larger number of well-dispersed amide signals appear in the spectrum of the CNBD-bound form (*Figure 1C*). Importantly, we were now able to assign the backbone chemical shift resonances of TRIP8b$_{nano}$ bound to the CNBD. The φ and ψ dihedral angles obtained from the NMR assignment indicate that the peptide displays two α-helices (stretch L$_{238}$-E$_{250}$ named helix N and stretch T$_{253}$-R$_{269}$ named helix C) when bound to CNBD. The helices are separated by two amino acids; three and six residues at the N- and C- termini, respectively, are unstructured (*Figure 1D*).

## Structural characterization of CNBD bound to TRIP8b$_{nano}$

NMR-analysis of the CNBD fragment bound to TRIP8b$_{nano}$ revealed that the interaction with the peptide does not affect the overall fold of the protein. Thus, the CNBD adopts the typical fold of the cAMP-free state, in line with previous evidence that this is the form bound by TRIP8b (*Saponaro et al., 2014*; *DeBerg et al., 2015*) More specifically, the secondary structure elements of the cAMP-free CNBD are all conserved in the TRIP8b$_{nano}$–bound CNBD (*Figure 2*). This finding generally agrees with a previous double electron-electron resonance (DEER) analysis of the CNBD - TRIP8b interaction, which showed that TRIP8b binds to a conformation largely similar to the cAMP-free state (*DeBerg et al., 2015*). Despite the overall agreement with the DEER study, the NMR data also reveal a new and unexpected feature of TRIP8b binding to the CNBD. Indeed, our results show that TRIP8b$_{nano}$ binding to the CNBD induces a well-defined secondary structure of the distal region of the C-helix (*Figure 2*). This means that the distal region of the C-helix (residues 657–662), which is unstructured in the free form of the CNBD (*Saponaro et al., 2014*; *Lee and MacKinnon, 2017*), extends into a helical structure upon ligand binding irrespectively of whether the ligand is cAMP (*Puljung and Zagotta, 2013*; *Saponaro et al., 2014*; *Lee and MacKinnon, 2017*) or TRIP8b (*Figure 2*). In contrast, and very differently from cAMP, which directly contacts the P-helix in the Phosphate Binding cassette (PBC) and causes its folding (*Saponaro et al., 2014*; *Lee and MacKinnon, 2017*), the NMR data show that TRIP8b$_{nano}$ binding to the CNBD does not induce P-helix formation (*Figure 2*).

## Modeling the CNBD - TRIP8b$_{nano}$ complex

Despite the significant improvement in sample stability and NMR spectra quality achieved upon TRIP8b$_{nano}$ binding, we were still unable to assign the side chains of both proteins in the complex and thus could not solve the solution structure of the complex by the canonical NMR procedure. We therefore built a model (*Figure 3*) of the CNBD - TRIP8b$_{nano}$ complex by docking the two NMR-derived structures described above using the Haddock program (a detailed description of how the respective structures were generated is provided in Materials and methods and *Figure 3—source data 1*).

In order to define the active residues (ambiguous interaction restraints) on the CNBD we used the chemical shift perturbation values as described in *Figure 3—figure supplement 1*. For TRIP8b$_{nano}$, we defined as active a stretch of residues, E$_{239}$-E$_{243}$, previously identified as critical for the interaction (*Santoro et al., 2011*). Output clusters of this first molecular docking calculation (settings can

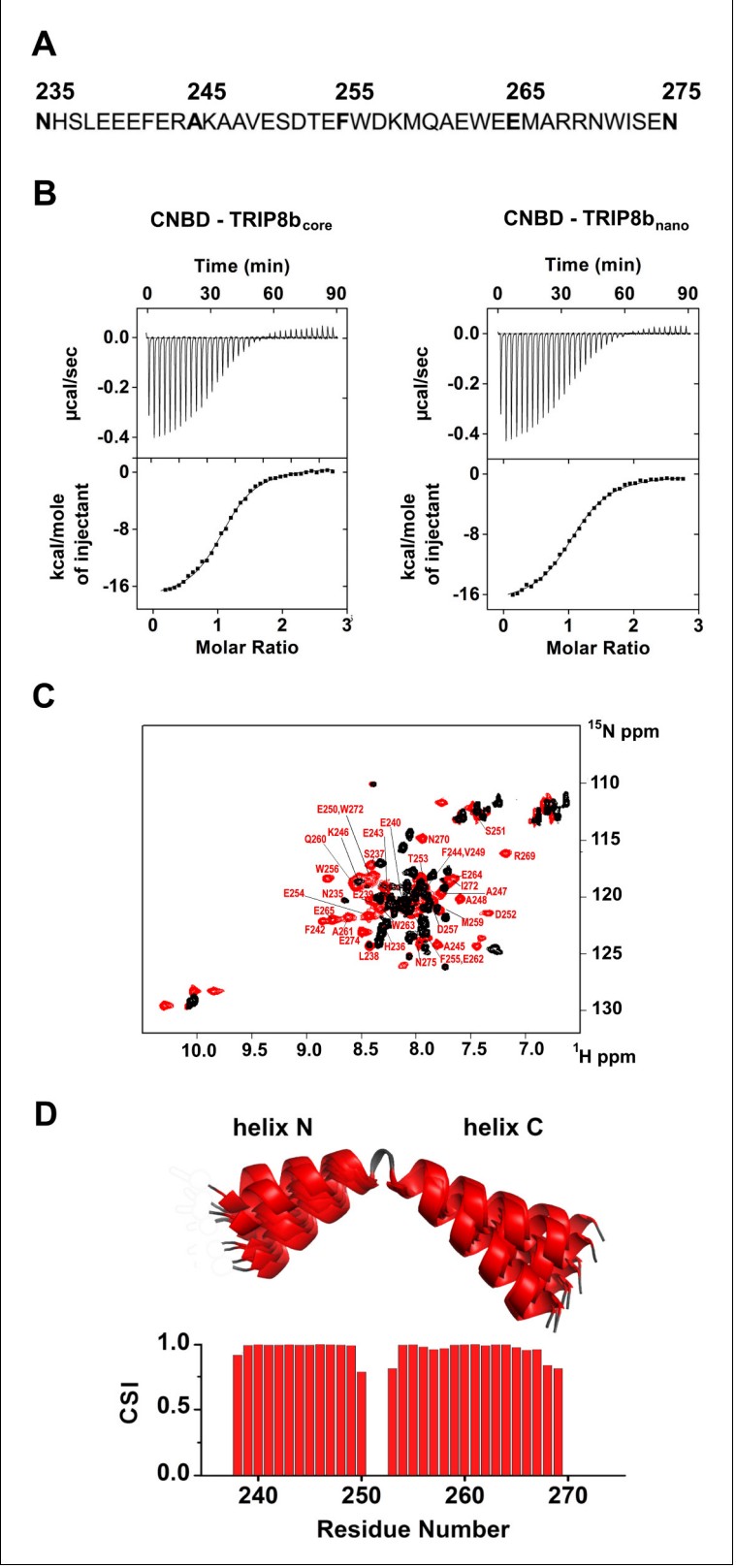

**Figure 1.** Functional and structural characterization of TRIP8b_nano. (**A**) Primary sequence of TRIP8b_nano. Amino acid numbering refers to full length mouse TRIP8b (1a4). (**B**) Binding of TRIP8b_core and TRIP8b_nano to purified His6-MBP-CNBD measured by Isothermal titration calorimetry (ITC). Upper panel, heat changes (μcal/sec) during successive injections of 8 μL of the corresponding TRIP8b peptide (200 μM) into the chamber containing His6-MBP-CNBD (20

*Figure 1 continued on next page*

*Figure 1 continued*

µM). Lower panel, binding curve obtained from data displayed in the upper panel. The peaks were integrated, normalized to TRIP8b peptide concentration, and plotted against the molar ratio (TRIP8b peptide/His$_6$-MBP-CNBD). Solid line represents a nonlinear least-squares fit to a single-site binding model, yielding, in the present examples, a $K_D$ = 1.2 ± 0.1 µM for TRIP8b$_{core}$ and $K_D$ = 1.4 ± 0.1 µM for TRIP8b$_{nano}$. (C) Evidence for TRIP8b$_{nano}$ folding upon CNBD binding based on the superimposition of the [$^1$H, $^{15}$N] heteronuclear single quantum coherence (HSQC) NMR spectrum of CNBD-free TRIP8b$_{nano}$ (black) and CNBD-bound TRIP8b$_{nano}$ (red). The latter experiment was performed at the molar ratio ([CNBD]/[TRIP8b$_{nano}$])=3. The backbone amide (HN) signals of the residues of CNBD-bound TRIP8b$_{nano}$ are labelled in red. (D) (Top) Ribbon representation of the 10 lowest energy conformers of TRIP8b$_{nano}$ bound to CNBD used for in silico modelling of CNBD-TRIP8b$_{nano}$ complex. The unfolded regions at the N- and C-termini of the construct (residues 235–237 and 270–275) are omitted for clarity. (Bottom) Chemical Shift Index (CSI, calculated using TALOS+) plotted as a function of the residue number of TRIP8b$_{nano}$ bound to CNBD. Positive values represent helical propensity.

DOI: https://doi.org/10.7554/eLife.35753.002

be found in Materials and methods) were further screened for TRIP8b$_{nano}$ orientations in agreement with a previous DEER analysis, which placed TRIP8b residue A$_{248}$ closer to the proximal portion and TRIP8b residue A$_{261}$ closer to the distal portion of the CNBD C-helix (*DeBerg et al., 2015*). Remarkably, in all clusters selected in this way, residues E$_{264}$ or E$_{265}$ in TRIP8b were found to interact with residues K$_{665}$ or K$_{666}$ of the CNBD (*Figure 3—figure supplement 2*). This finding was notable, because we previously identified K$_{665}$/K$_{666}$ as being critical for TRIP8b interaction in a biochemical binding assay (*Saponaro et al., 2014*). We thus proceeded to individually mutate each of these four positions, and test their effect on binding affinity through ITC. As expected, reverse charge mutations K$_{665}$E or K$_{666}$E (CNBD) as well as E$_{264}$K or E$_{265}$K (TRIP8b$_{nano}$) each strongly reduced the CNBD/TRIP8b$_{nano}$ binding affinity (*Figure 3—figure supplement 3*).

Based on these observations, we performed a second molecular docking calculation, including E$_{264}$ and E$_{265}$ as additional active residues for TRIP8b$_{nano}$. This procedure resulted in the model shown in *Figure 3*, which represents the top-ranking cluster for energetic and scoring function (*Figure 3—source data 2*) and was fully validated by mutagenesis analysis as described below. Scrutiny of the model shows that TRIP8b$_{nano}$ binds to both the C-helix and the N-bundle loop (*Figure 3A*). Binding to the C-helix is mainly guided by electrostatic interactions between the negative charges on TRIP8b$_{nano}$, and the positive charges on the CNBD (*Figure 3A*). As shown in *Figure 3B*, the model highlights a double saline bridge (K$_{665}$ and K$_{666}$ of CNBD with E$_{265}$ and E$_{264}$ of TRIP8b$_{nano}$) in line with the ITC results described above (*Figure 3—figure supplement 3*). Of note, the contribution of residue R$_{662}$ to the binding is also consistent with previous experiments showing residual TRIP8b interaction in a CNBD deletion mutant ending at position 663 (*Saponaro et al., 2014*). Our modeling data suggest that, upon folding of the distal portion of the C-helix, the side chains of residues R$_{662}$ and K$_{665}$ face to the inside when contacting cAMP, but face to the outside when binding TRIP8b (*Figure 3—figure supplement 4*). This indicates that cAMP and TRIP8b directly compete for the binding to the distal region of C-helix.

In addition to clarifying the role of residues in the distal portion of the CNBD C-helix, the model also highlights a second important cluster of electrostatic interactions, with R$_{650}$ in the proximal portion of the CNBD C-helix contacting E$_{240}$ and E$_{241}$ in helix N of TRIP8b$_{nano}$ (*Figure 3C*). To confirm the contribution of these residues, we reversed charges and tested each residue mutation for binding in ITC. The results in *Figure 3—figure supplement 3* show that R$_{650}$E caused a more than six-fold reduction in binding affinity for TRIP8b$_{nano}$, with smaller but significant effects seen also for E$_{240}$R and E$_{241}$R.

A third important contact highlighted by the model is the interaction between N$_{547}$ in the N-bundle loop of the CNBD and D$_{252}$ in the link between helix N and helix C of TRIP8b$_{nano}$ (*Figure 3D*). We tested this potential interaction by disrupting the expected hydrogen bond between N$_{547}$ and the carboxyl group of the negative residue (D$_{252}$) in TRIP8b$_{nano}$. The asparagine in CNBD was mutated into aspartate (N$_{547}$D) to generate an electrostatic repulsion for D$_{252}$, and the carboxyl group in D$_{252}$ of TRIP8b$_{nano}$ was removed by mutation into asparagine (D$_{252}$N). As predicted, N$_{547}$D greatly reduced binding to TRIP8b in ITC assays (*Figure 3—figure supplement 3*), with a smaller but significant effect observed also for D$_{252}$N (*Figure 3—figure supplement 3*). These results

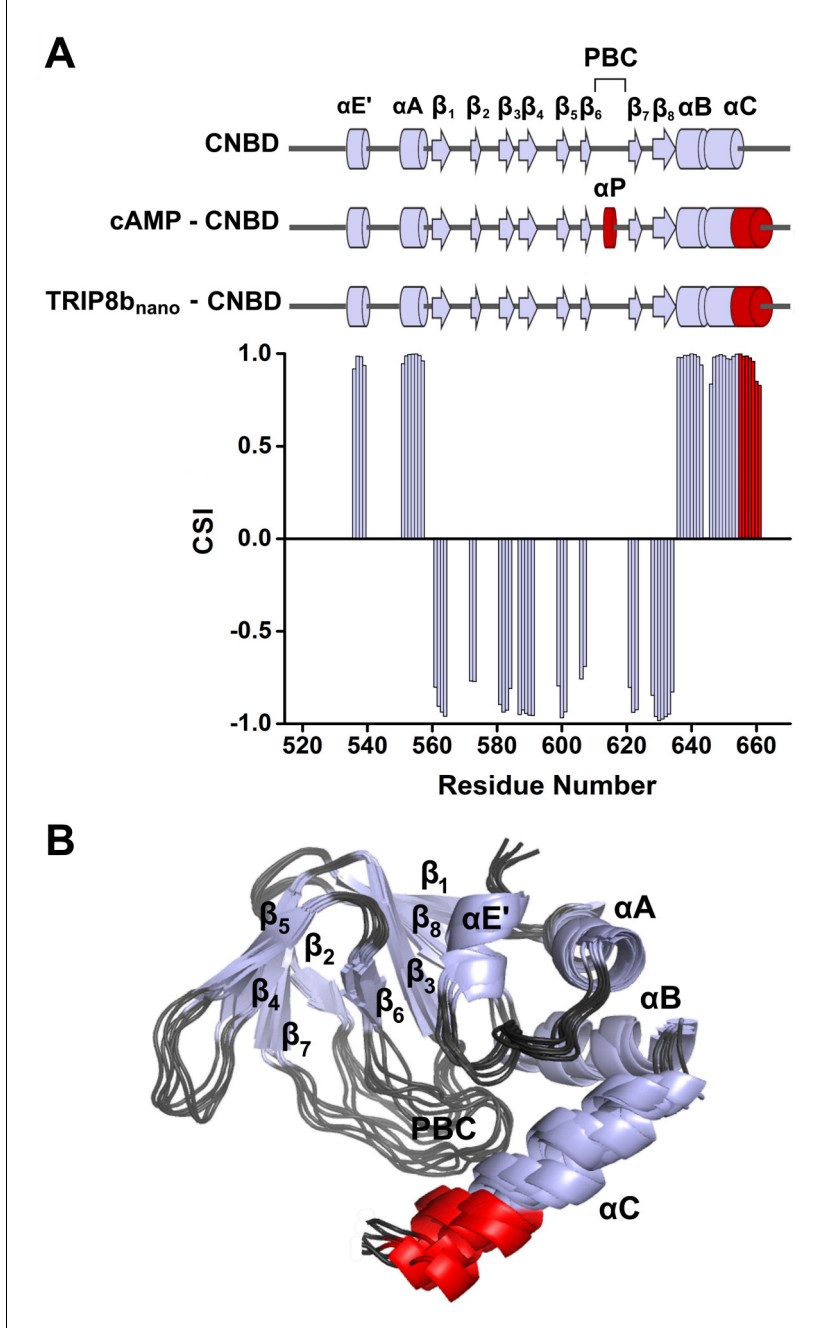

**Figure 2.** NMR structure of CNBD bound to TRIP8b$_{nano}$. (**A**) (Top) comparison of secondary structure elements of cAMP-free CNBD (*Saponaro et al., 2014*), cAMP-bound CNBD (*Zagotta et al., 2003*) and cAMP-free CNBD bound to TRIP8b$_{nano}$ (this study). Secondary structure elements are indicated by arrows (β-strands) and cylinders (α-helices) and labeled. The loop between β$_6$ and β$_7$ constitutes the Phosphate Binding Cassette (PBC). The elements that fold upon binding of cAMP and TRIP8b$_{nano}$ are shown in red. (Bottom) Chemical Shift Index (CSI, calculated using TALOS+) plotted as a function of the residue number of CNBD bound to TRIP8b$_{nano}$. Positive values represent helical propensity, while negative values represent strands. (**B**) Ribbon representation of the 10 lowest energy conformers of CNBD bound to TRIP8b$_{nano}$ used for in silico modeling of CNBD - TRIP8b$_{nano}$ complex. Secondary structure elements are coloured in light gray and labeled. Loop regions are colored in dark gray. The distal region of the C-helix (residues 657–662), which is unfolded in the free form of the CNBD (*Saponaro et al., 2014*) and folds upon TRIP8b$_{nano}$ binding, is coloured in red. The unfolded regions at the N- and C-termini of the construct (residues 521–532 and 663–672 respectively) are omitted for clarity.

DOI: https://doi.org/10.7554/eLife.35753.003

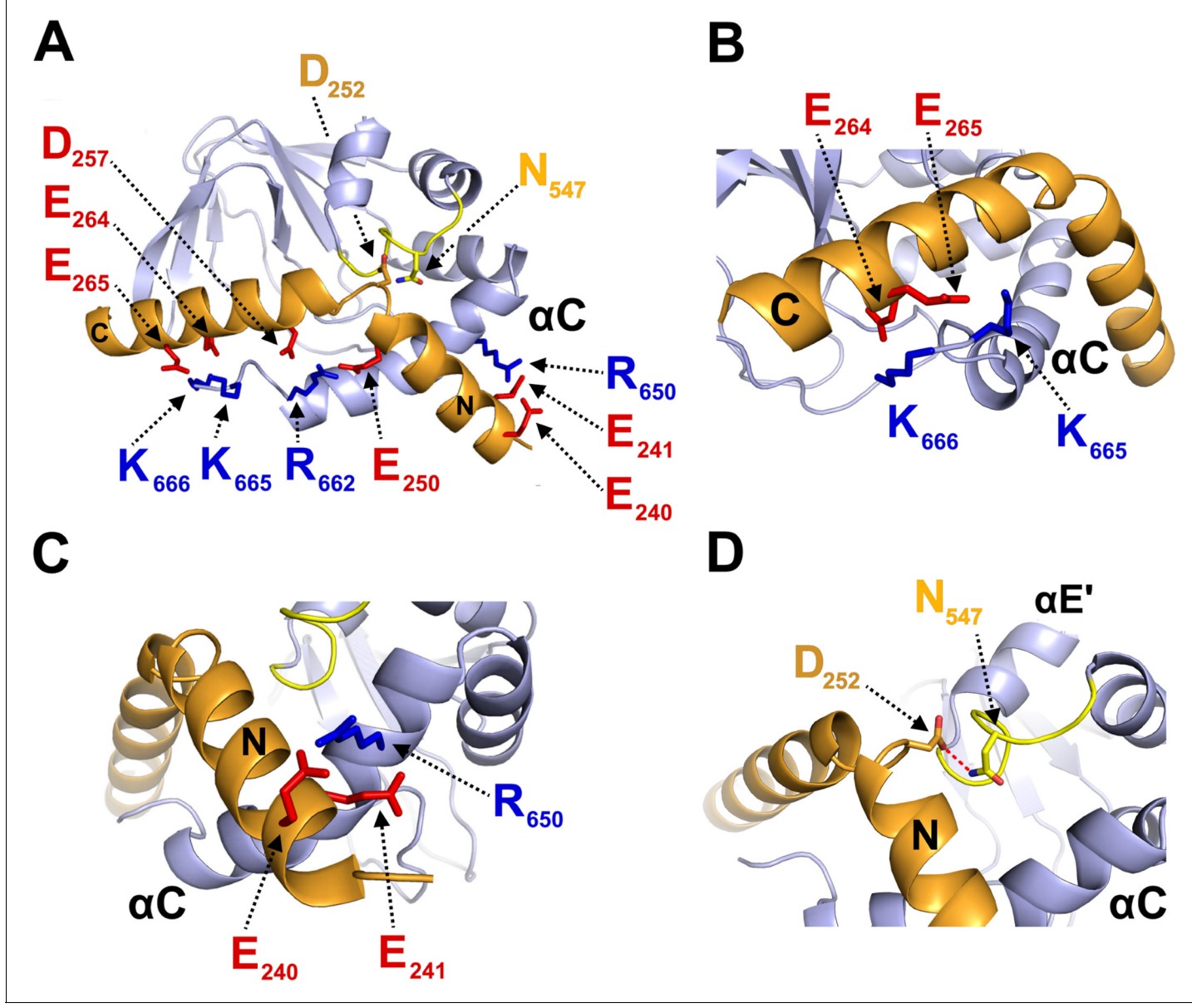

**Figure 3.** Structural model of CNBD – TRIP8b$_{nano}$ complex. (**A**) Ribbon representation of the complex where CNBD is in gray and TRIP8b$_{nano}$ is in orange. Helix N (**N**) and helix C (**C**) of TRIP8b$_{nano}$ are labeled. C-helix of CNBD ($\alpha$C) is labeled, while N-bundle loop is colored in yellow. Positively charged residues of C-helix CNBD (blue) and negatively charged residues of TRIP8b$_{nano}$ (red) involved in salt bridges are shown as sticks and labeled. N$_{547}$ of the N-bundle loop (yellow) and D$_{252}$ of TRIP8b$_{nano}$ (orange) are shown as sticks and labeled. (**B**) Close view of K$_{665}$ and K$_{666}$ of CNBD that interact, respectively, with E$_{265}$ and E$_{264}$ of TRIP8b$_{nano}$. C-helix ($\alpha$C) of CNBD, and Helix C (**C**) of TRIP8b$_{nano}$ are labeled. (**C**) Close view of R$_{650}$ of CNBD that is positioned between E$_{240}$ and E$_{241}$ of TRIP8b$_{nano}$. C-helix ($\alpha$C) of CNBD, and Helix N (**N**) of TRIP8b$_{nano}$ are labeled. (**D**) Close view of N$_{547}$ of N-bundle loop that forms a hydrogen bond (red dashed line) with D$_{252}$ of TRIP8b$_{nano}$. Helix E' ($\alpha$E') and C-helix ($\alpha$C) of CNBD, and Helix N (**N**) of TRIP8b$_{nano}$ are labeled.

DOI: https://doi.org/10.7554/eLife.35753.004

The following source data and figure supplements are available for figure 3:

**Source data 1.** Acquisition parameters for NMR experiments performed on cAMP-free human HCN2 CNBD in complex with TRIP8b$_{nano}$ and vice-versa.
DOI: https://doi.org/10.7554/eLife.35753.011
**Source data 2.** Docking calculation.
DOI: https://doi.org/10.7554/eLife.35753.012
**Figure supplement 1.** CNBD residues involved in TRIP8b$_{nano}$ binding.
DOI: https://doi.org/10.7554/eLife.35753.005
**Figure supplement 2.** Representative families of clusters obtained from the first docking calculation.

*Figure 3 continued on next page*

*Figure 3 continued*

DOI: https://doi.org/10.7554/eLife.35753.006

**Figure supplement 3.** Biochemical validation of CNBD – TRIP8b$_{nano}$ complex.

DOI: https://doi.org/10.7554/eLife.35753.007

**Figure supplement 4.** Different orientation of R$_{662}$ and K$_{665}$ in the cAMP-bound and TRIP8b$_{nano}$-bound conformation of the CNBD.

DOI: https://doi.org/10.7554/eLife.35753.008

**Figure supplement 5.** Mutation N$_{520}$D affects cAMP affinity in full-length HCN2 channel.

DOI: https://doi.org/10.7554/eLife.35753.009

**Figure supplement 6.** Structural characterization of N$_{547}$D CNBD protein.

DOI: https://doi.org/10.7554/eLife.35753.010

confirm and extend our previous finding that the N-bundle loop contributes in a substantial manner to the binding of TRIP8b (*Saponaro et al., 2014*). To understand the mechanism for the allosteric effect of TRIP8b on cAMP binding, which has been postulated on the basis of electrophysiological and structural data (*Hu et al., 2013*; *Saponaro et al., 2014*), we further tested by ITC the affinity of the N$_{547}$D CNBD mutant for cAMP. Somewhat surprisingly, the affinity of the mutant is much lower than that of the wt (N$_{547}$D K$_D$ = 5.5 ± 0.4 μM (n = 3) *vs.* wt K$_D$ = 1.4 ± 0.1 μM (n = 3)). Moreover, we measured a reduced sensitivity to cAMP also in patch clamp experiments where addition of 5 μM cAMP caused a right shift in the V$_{1/2}$ of the mutant HCN2 channel of only 5 mV while the wt channel shifted by 12 mV (*Figure 3—figure supplement 5*). To exclude that the N$_{547}$D mutation affects the overall structure of the CNBD, we performed the NMR ($^1$H-$^{15}$N HSQC spectrum) analysis of the N$_{547}$D CNBD mutant. Our data show that the protein is appropriately folded (*Figure 3—figure supplement 6*). In conclusion, since the N-bundle loop does not directly contact any of the residues of the cAMP binding pocket, these findings underscore a previously unaddressed role of the N-bundle loop in allosterically modulating cAMP binding to the CNBD (see Discussion).

## TRIP8b$_{nano}$ as a tool for the direct regulation of native HCN currents

Next, we asked whether the relatively short TRIP8b$_{nano}$ could be used to block cAMP-dependent modulation of HCN channels by delivering the peptide to full length channels. To this end, we dialyzed TRIP8b$_{nano}$ into the cytosol of HEK 293 T cells transfected either with HCN1, HCN2, or HCN4 channels. The peptide was added (10 μM) in the recording pipette together with a non-saturating concentration of cAMP (5 μM for HCN2, 1 μM for HCN4) expected to induce a ~ 10 mV rightward shift in the half-activation potential (V$_{1/2}$) of the channels (*Figure 4*). No cAMP was added in the case of HCN1, because, in HEK 293 T cells, this isoform is already fully shifted to the right by the endogenous cAMP and does not respond further (*Figure 4—figure supplement 1*). Indeed, it is possible to induce a ~10 mV left shift in HCN1 V$_{1/2}$ by introducing the mutation R$_{549}$E that prevents cAMP binding to the CNBD (*Figure 4—figure supplement 1*).

*Figure 4A–C* show representative current traces recorded at four given voltages, in control solution, cAMP, and cAMP +10 μM TRIP8b$_{nano}$ (HCN4 and HCN2) or +10 μM TRIP8b$_{nano}$ only (HCN1) in the patch pipette. Already from a visual comparison of the most positive voltage at which the current appears measurable, it is evident that TRIP8b$_{nano}$ counteracts the activating effect of cAMP on the voltage-dependent gating. In the case of HCN1, the effect of TRIP8b$_{nano}$ can be observed without added cAMP for the aforementioned reasons. *Figure 4D–F* show the mean channel activation curves obtained from the above and other experiments. Fitting the Boltzmann equation to the data (solid and dashed lines of *Figure 4D–F*, see Materials and methods for equation) yielded the half-activation potential values (V$_{1/2}$) plotted in *Figure 4G*. The addition of TRIP8b$_{nano}$ prevents the cAMP-induced right shift of about 13 mV in HCN4 (V$_{1/2}$ = -102.8, –89.2, –102.1 mV, for control, cAMP and cAMP + TRIP8b$_{nano}$, respectively), and of about 11 mV in HCN2 (V$_{1/2}$ = –93.7, –83.5, –94.5 mV, for control, cAMP and cAMP + TRIP8b$_{nano}$, respectively). In HCN1, TRIP8b$_{nano}$ induced a left shift in V$_{1/2}$ of about 10 mV (from – 72.8 to –82 mV) which is comparable to that induced by the R$_{549}$E mutation (from –72.7 to –80.4 mV) (*Figure 4—figure supplement 1*).

*Figure 4G* also shows the result of a control experiment performed on HCN4 where 10 μM TRIP8b$_{nano}$ was added to the extracellular medium (TRIP8b$_{nano}$ bath) in order to test if the peptide was able to cross the cell membrane (current traces and activation curves not shown). The V$_{1/2}$ value, which is similar to that of cAMP alone (–91.7 vs. –89.2 mV), confirmed that TRIP8b$_{nano}$ peptide

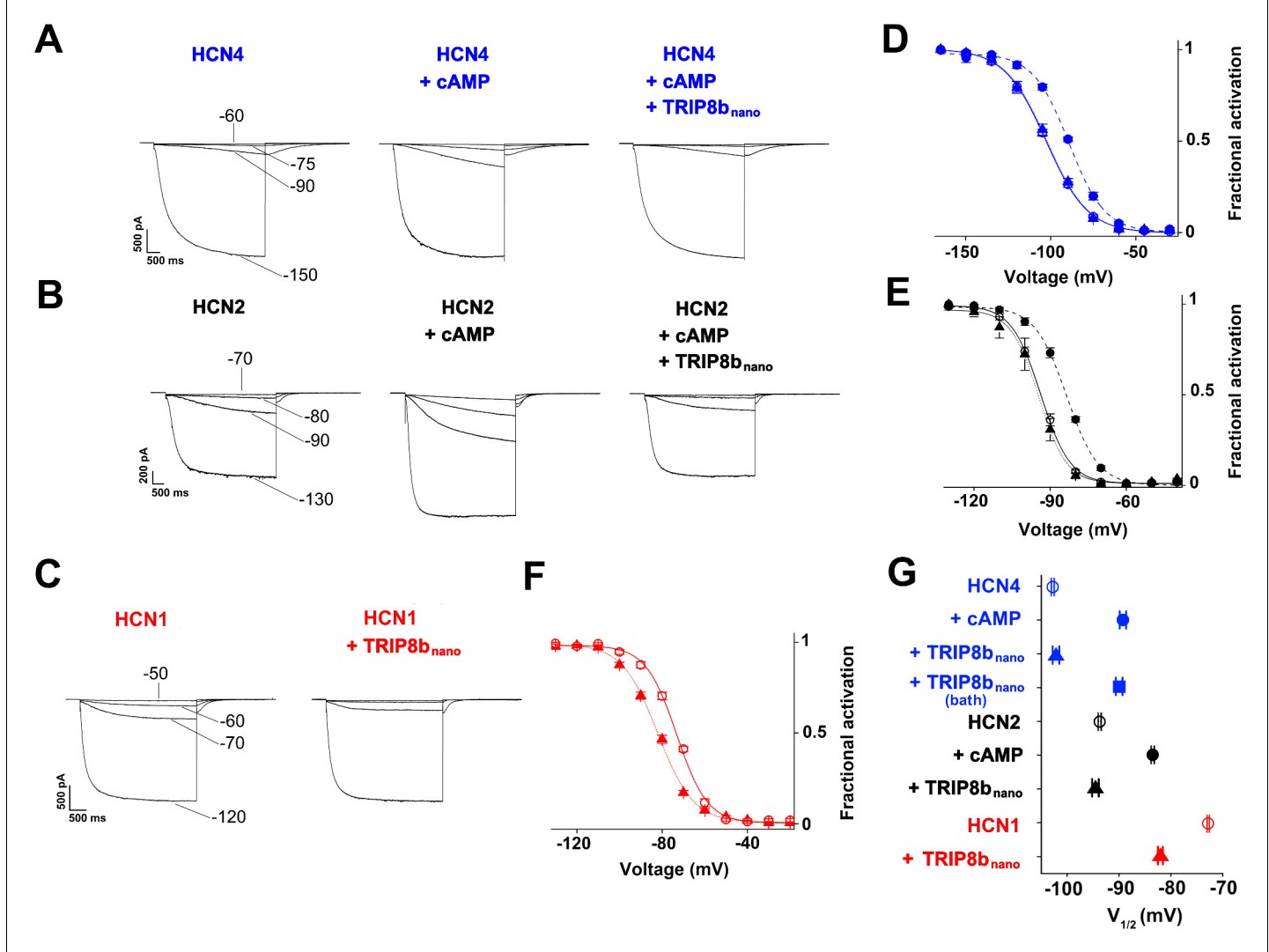

**Figure 4.** TRIP8b$_{nano}$ abolishes cAMP effect on HCN channel gating. (**A-C**) Representative whole-cell HCN4, HCN2 and HCN1 currents recorded, at the indicated voltages, with control solution or with cAMP (1 µM for HCN4 and 5 µM for HCN2) or with cAMP + 10 µM TRIP8b$_{nano}$ in the patch pipette (for HCN1, 10 µM TRIP8b$_{nano}$ only was added). (**D-F**) Mean activation curves measured from HCN4, HCN2 and HCN1 in control solution (open circles), cAMP (filled circles), cAMP +TRIP8b$_{nano}$, or TRIP8b$_{nano}$ only in the case of HCN1 (filled triangles). Solid and dashed lines indicate Boltzmann fitting to the data (see Materials and methods). (**G**) Half activation potential (V$_{1/2}$) values of HCN4 (blue), HCN2 (black) HCN1 (red) in control solution (open circle), cAMP (filled circle) and cAMP +TRIP8b$_{nano}$, or TRIP8b$_{nano}$ only in the case of HCN1 (filled triangle). HCN4, control = −102.8 ± 0.3 mV; HCN4 +1 µM cAMP = −89.2 ± 0.6 mV; HCN4 +1 µM cAMP +10 µM TRIP8b$_{nano}$ = −102.1 ± 0.6 mV, HCN4 +1 µM cAMP in the patch pipette +10 µM TRIP8b$_{nano}$ in the bath solution = −91.7 ± 0.3 mV; HCN2, control = −93.7 ± 0.3 mV; HCN2 +5 µM cAMP = −83.5 ± 0.3 mV; HCN2 +5 µM cAMP +10 µM TRIP8b$_{nano}$ = −94.5 ± 0.6 mV; HCN1, control = −72.8 ± 0.2 mV; HCN1 +10 µM TRIP8b$_{nano}$ = −82 ± 0.5 mV. Data are presented as mean ± SEM. Number of cells (N) ≥ 11. There is no significant difference between the controls and the addition of TRIP8b$_{nano}$ with (HCN4, HCN2) or without (HCN1) cAMP in the pipette. No significant difference was observed following the addition of TRIP8b$_{nano}$ in the bath. Statistical analysis performed with two-way ANOVA, followed by post-hoc Tukey test (p<0.001).

DOI: https://doi.org/10.7554/eLife.35753.013

The following figure supplements are available for figure 4:

**Figure supplement 1.** Comparison of half activation potentials (V$_{1/2}$) of HCN1 WT and HCN1 R$_{549}$E mutant (*Chen et al., 2001*).

DOI: https://doi.org/10.7554/eLife.35753.014

**Figure supplement 2.** Voltage-dependency of activation time constant (τ$_{on}$) of HCN4 channels in control solution (open circles), 1 µM cAMP (filled circles), 1 µM cAMP + 10 µM TRIP8b$_{nano}$ (filled triangles).

DOI: https://doi.org/10.7554/eLife.35753.015

affects channel gating only if added to the intracellular solution presumably because it does not diffuse through the cell membrane.

It is worth noting that TRIP8b$_{nano}$ prevents other related effects of cAMP activation in HCN channels, such as the acceleration of activation kinetics (*Wainger et al., 2001*) and, for HCN2 only, the increase in maximal current (*Chen et al., 2007*; *Hu et al., 2013*). For example, the activation kinetics ($\tau_{on}$) of HCN4 measured at −120 mV was: control = 2 ± 0.2 s, 1 μM cAMP = 1.2 ± 0.1 s, 1 μM cAMP + 10 μM TRIP8b$_{nano}$ = 2 ± 0.3 s (*Figure 4—figure supplement 2*). Moreover, *Figure 4B* clearly shows that TRIP8b$_{nano}$ fully prevented the increase in maximal current in HCN2.

Based on these results, we reckoned the peptide may be employed as a regulatory tool for native I$_f$/I$_h$ currents. As proof of principle, we tested whether TRIP8b$_{nano}$ can modulate the frequency of action potential firing in SAN myocytes. In these cells, I$_f$ is key contributor of the diastolic depolarization phase of the pacemaker action potential cycle. Moreover, the autonomic nervous system modulates the frequency of action potential firing by changing intracellular cAMP levels, which in turn acts on f-HCN channel open probability (*DiFrancesco, 1993*). We thus recorded the native I$_f$ current in acutely isolated rabbit SAN myocytes with and without 10 μM TRIP8b$_{nano}$ in the pipette solution (*Figure 5A*). *Figure 5B* shows that the averaged I$_f$ activation curve measured in presence of TRIP8b$_{nano}$ is significantly shifted to hyperpolarized voltages compared to the control. This indicates that the peptide is displacing the binding of endogenous cAMP to native HCN channels. Moreover, when the experiment was repeated in the presence of 1 μM cAMP, TRIP8b$_{nano}$ prevented the typical cAMP-dependent potentiation of the native I$_f$ current (*Figure 5B*). In light of these results, we tested whether TRIP8b$_{nano}$ is also able to modulate cardiac automaticity by antagonizing basal cAMP. The data in *Figure 5C* show that TRIP8b$_{nano}$ indeed significantly decreased the rate of action potential firing in single SAN cells. Strikingly, the observed 30% decrease in action potential rate corresponds to the effect induced by physiological concentrations of acetylcholine (*DiFrancesco et al., 1989*).

To conclusively prove that the inhibition of the native I$_f$ current was specifically due to TRIP8b$_{nano}$ rather than caused by the dilution of the cellular content following whole cell configuration, we created a TAT version of TRIP8b$_{nano}$ (hereafter TAT-TRIP8b$_{nano}$). Indeed, the TAT sequence allows the entry of biomolecules into a cell via endocytosis and/or direct translocation across the plasma membrane, thus leaving the cytosolic content unaltered (*Guidotti et al., 2017*).

We therefore tested whether both TRIP8b$_{nano}$ and TAT-TRIP8b$_{nano}$ were able to selectively inhibit the β-adrenergic stimulation of I$_f$ current, while leaving the potentiation of L-type Ca$^{2+}$ current (I$_{Ca,L}$) unaltered. To this end, we recorded either the native I$_f$ or I$_{Ca,L}$ current from cardiomyocytes acutely isolated from mouse sinoatrial node (SAN) in the presence and in the absence of 10 μM TRIP8b$_{nano}$ or TAT-TRIP8b$_{nano}$, before and after stimulation with 100 nM isoproterenol (ISO), a β-adrenergic receptor agonist (*Figure 6*). Strikingly, TRIP8b$_{nano}$ prevented the isoproterenol-induced increase of I$_f$ current density, both when the peptide was added in the recording pipette solution (*Figure 6A and B*), and when it was used in the TAT version added to the bath (*Figure 6A and C*). The specificity of TRIP8b$_{nano}$ for I$_f$ current was confirmed by the absent inhibition of basal I$_{Ca,L}$ (*Figure 6D*). In addition, we failed to record a significant difference in the isoproterenol-stimulated increase of the I$_{CaL}$ current density between the control condition and 10 μM TRIP8b$_{nano}$ (*Figure 6E*) or TAT-TRIP8b$_{nano}$ (*Figure 6F*) conditions. To test whether the TAT-TRIP8b$_{nano}$ effect described above was exclusively due to TRIP8b$_{nano}$ peptide, we repeated the experiments with a scrambled version of the peptide (TAT- (SCRAMBLED) TRIP8b$_{nano}$) to exclude that the effect could be due to the TAT sequence (*Figure 6—figure supplement 1*). We failed to observe a significant reduction in the responsiveness of I$_f$ to isoproterenol in the presence of TAT- (SCRAMBLED) TRIP8b$_{nano}$ confirming that prevention of cAMP induced f- current stimulation was specific of the TRIP8b$_{nano}$ sequence.

## Discussion

### TRIP8b-CNBD complex

In this study, we have identified the minimal binding peptide that reproduces the effects of TRIP8b on HCN channel gating. The peptide is 40 aa long and binds the HCN CNBD with high affinity ($K_D$ = 1.4 μM). By solving the NMR structures of TRIP8b$_{nano}$ and HCN CNBD in the bound form, we generated a structural model of their complex. The model provides detailed information on this protein-protein interaction at the atomic level with implications on their physiological function. The data

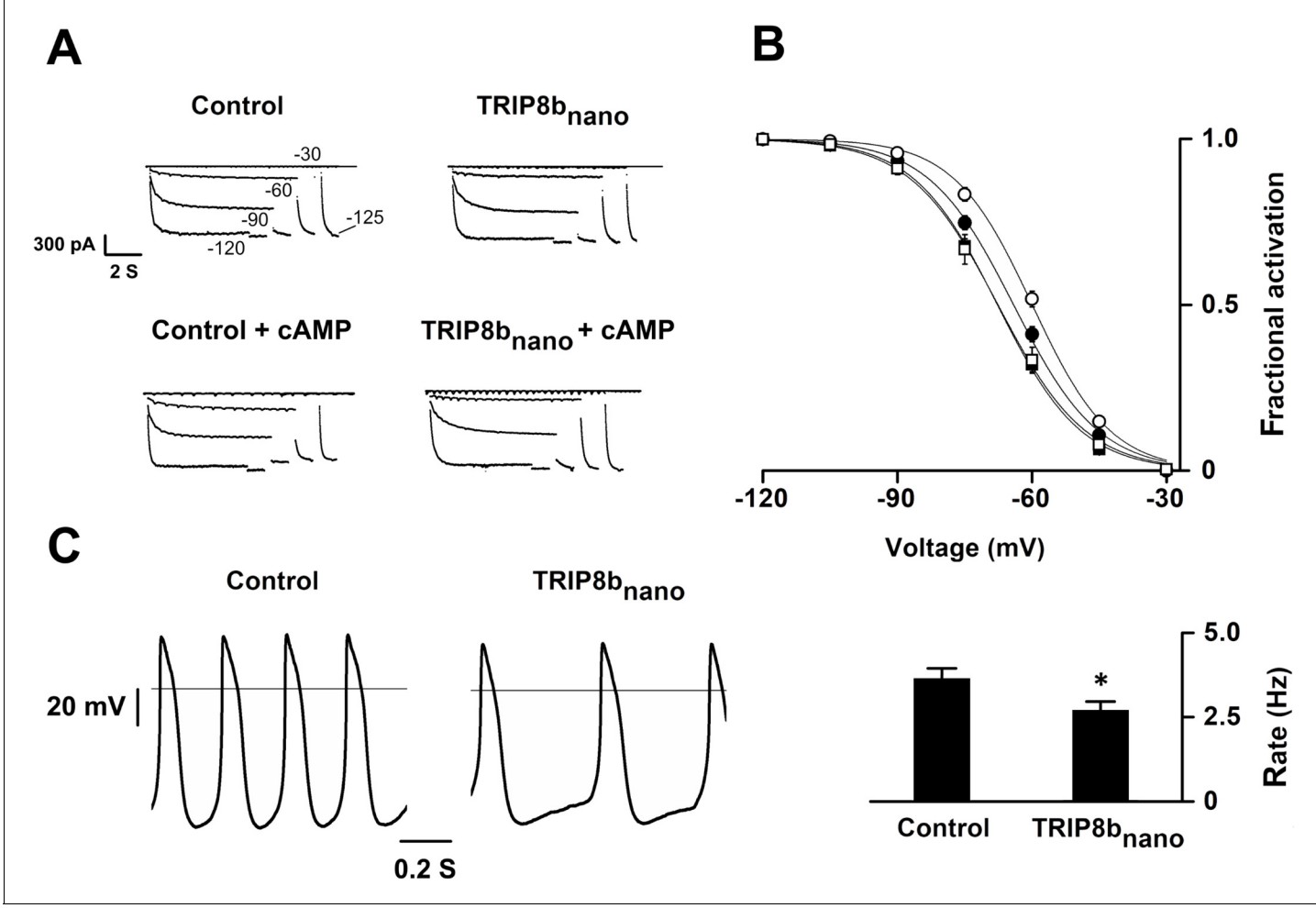

**Figure 5.** Effects of TRIP8b_nano on voltage-dependent activation of $I_f$ and spontaneous rate in rabbit sinoatrial node (SAN) myocytes. (**A**) Representative whole-cell $I_f$ currents recorded, at the indicated voltages, in the control solution and in the presence of 10 μM TRIP8b_nano, without (top) and with 1 μM cAMP in the pipette (bottom). (**B**) Mean $I_f$ activation curves were measured using a two-step protocol (see Materials and methods) in control (filled circles) or in the presence of: 1 μM cAMP (open circles); 10 μM TRIP8b_nano (filled squares); 1 μM cAMP +10 μM TRIP8b_nano (open squares). Ligands were added in the patch pipette. Half activation potential ($V_{1/2}$) of $I_f$ activation curves measured in control = −64.1 ± 0.4 mV or in the presence of: 1 μM cAMP = −59.9 ± 0.4 mV; 10 μM TRIP8b_nano = −67.7 ± 0.4 mV; 1 μM cAMP +10 μM TRIP8b_nano = −67.6 ± 0.7 mV. Data are presented as mean ± SEM. Number of cells (**N**) was ≥15. $V_{1/2}$ values are significantly different between each other's whit the exception of $V_{1/2}$ obtained in the presence TRIP8b_nano and cAMP +TRIP8 _bnano. Statistical analysis performed with two-way ANOVA, followed by post-hoc Bonferroni test (*p<0.05) (**C**) (Left) Representative recordings of single SAN cell spontaneous activity in control and in the presence of 10 μM TRIP8b_nano. (Right) Mean spontaneous rate (Hz) recorded in control solution = 3.65 ± 0.29 Hz and in the presence of 10 μM TRIP8b_nano added to the pipette = 2.69 ± 0.27 Hz. Data are presented as mean ± SEM. Number of cells (**N**) was ≥7. Statistical analysis performed with t test (*p<0.05).

DOI: https://doi.org/10.7554/eLife.35753.016

show that the minimal binding unit of TRIP8b, TRIP8b_nano, folds in two helices upon binding and suggest that this region is intrinsically disordered when it is not bound. The model structurally validates previous indirect evidence, which suggested that TRIP8b binds to two discrete elements of the CNBD: the N-bundle loop and the C-helix (*Saponaro et al., 2014*). The complex forms by electrostatic interactions, which are spread throughout the contact surface. As a consequence of the interaction with TRIP8b_nano, the C-helix of CNBD increases in length, a behavior previously observed in the case of cAMP binding (*Puljung and Zagotta, 2013*). This portion of C-helix includes the two residues $R_{662}$ and $K_{665}$ engaged in salt bridge formation with respectively $E_{250}$/$D_{257}$ and $E_{264}$ of TRIP8b_nano. It is important to note that these two cationic residues are also involved in cAMP binding (*Zagotta et al., 2003*; *Zhou and Siegelbaum, 2007*; *Lolicato et al., 2011*). The finding that TRIP8b and cAMP share the same binding sites on the C-helix provides a solid molecular explanation for

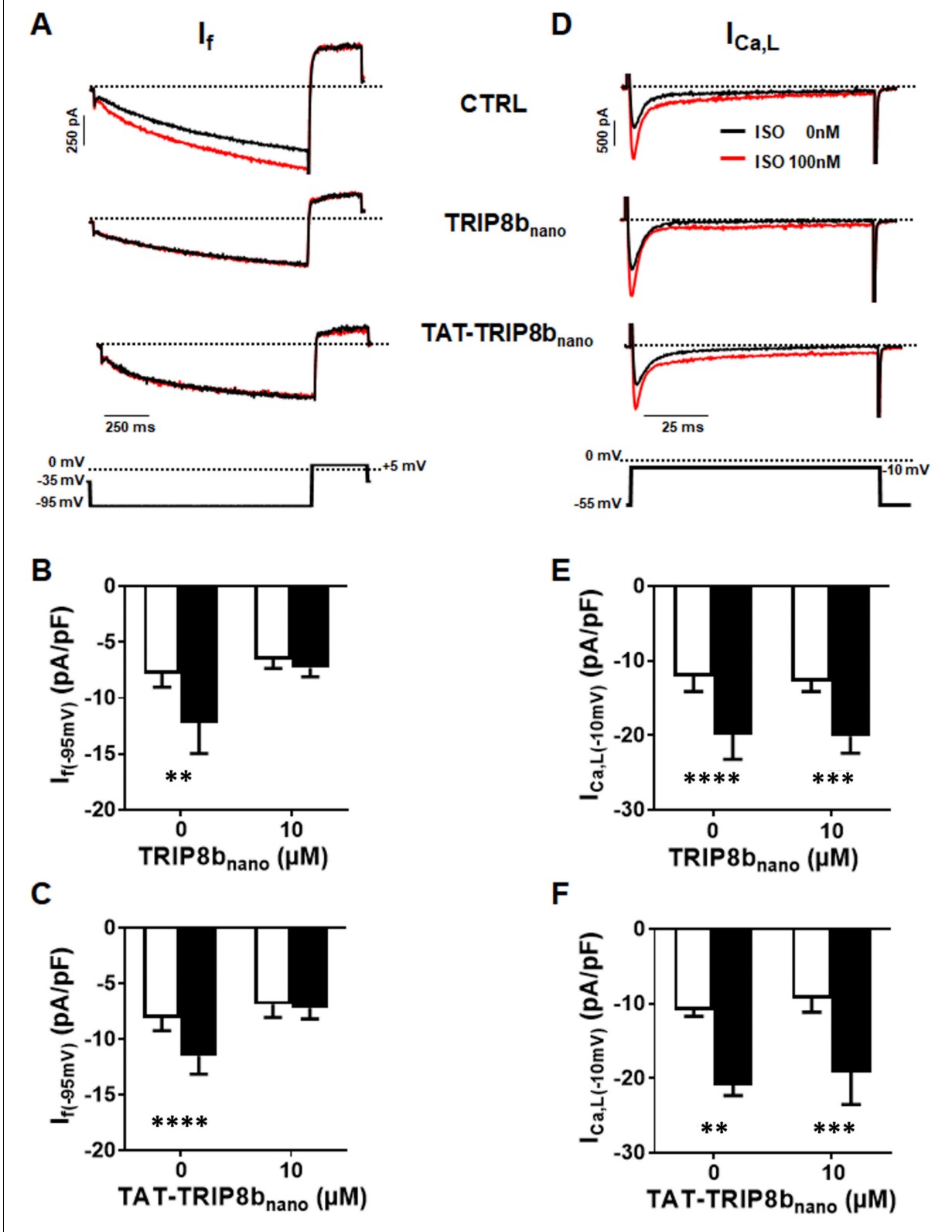

**Figure 6.** Effect of TRIP8b$_{nano}$ and TAT-TRIP8b$_{nano}$ on I$_f$ and I$_{Ca,L}$ in mouse sinoatrial node (SAN) myocytes. (A) Representative examples of I$_f$ recordings at −95 mV in control conditions (top), in 10 μM TRIP8b$_{nano}$ dialyzed cell (middle) and in cells perfused with 10 μM TAT-TRIP8b$_{nano}$ (bottom), before (black trace) and after (red trace) application of ISO 100 nM. The voltage-clamp protocol used for recordings is shown above current traces. (B) Mean normalized I$_f$ current density recorded at −95 mV in absence and in presence of 10 μM TRIP8b$_{nano}$ in the patch pipette, before (open bars) and after

*Figure 6 continued*

(filled bars) 100 nM ISO perfusion. Data are presented as mean ± SEM. Number of cells (N) ≥ 6. Statistical analysis performed with two-way ANOVA test, followed by Sidak multiple comparisons test (**p<0.01). (C) Mean normalised $I_f$ current density recorded at −95 mV in control solution or in the solution containing 10 µM TAT-TRIP8b$_{nano}$, in absence (open bars) and in the presence (filled bars) of 100 nM ISO. Data are presented as mean ± SEM. Number of cells (N) ≥ 8. Statistical analysis performed with two-way ANOVA test, followed by Sidak multiple comparisons test (****p<0.0001). (D) Representative examples of $I_{Ca,L}$ recordings at −10 mV in control conditions (top), in 10 µM TRIP8b$_{nano}$ dialyzed cell (middle) and in cells perfused with 10 µM TAT-TRIP8b$_{nano}$ (bottom), before (black trace) and after (red trace) application of ISO 100 nM. The voltage-clamp protocol used for recordings is shown above current traces. (E) Mean normalized $I_{Ca,L}$ current density recorded at −10 mV in absence and in presence of 10 µM TRIP8b$_{nano}$ in the patch pipette, before (open bars) and after (filled bars) 100 nM ISO perfusion. Data are presented as mean ± SEM. Number of cells (N) ≥ 8. Statistical analysis performed with two-way ANOVA test, followed by Sidak multiple comparisons test (***p<0.001; ****p<0.0001). (F) Mean normalized $I_{Ca,L}$ current density recorded at −10 mV in control solution or in the solution containing 10 µM TAT-TRIP8b$_{nano}$, in absence (open bars) and in the presence (filled bars) of 100 nM ISO. Data are presented as mean ± SEM. Number of cells (N) ≥ 7. Statistical analysis performed with two-way ANOVA test, followed by Sidak multiple comparison test (**p<0.01; ***p<0.001).

DOI: https://doi.org/10.7554/eLife.35753.017

The following figure supplement is available for figure 6:

**Figure supplement 1.** Effect of TAT- (SCRAMBLED) TRIP8b$_{nano}$ on $I_f$ in mouse sinoatrial node (SAN) myocytes.

DOI: https://doi.org/10.7554/eLife.35753.018

functional data, which imply a competition between the two regulators (*Han et al., 2011*; *DeBerg et al., 2015*; *Bankston et al., 2017*). Another study, however, has indicated that a direct competition model cannot fully explain the mutually antagonistic effect of the two ligands (*Hu et al., 2013*). Specifically, the fact that the inhibitory effect of TRIP8b on channel activity persists even at saturating cAMP concentrations advocated an allosteric component in the regulation mechanism. Our data, showing that the N$_{547}$D mutation in the N-bundle loop controls cAMP affinity in the binding pocket support the conclusion that the N-bundle loop allosterically controls cAMP binding. This is not surprising, given its crucial role of mechanically transducing to the pore the cAMP binding event within the CNBD (*Saponaro et al., 2014*).

The structural model also explains why a previously identified peptide selected by *Lyman et al. (2017)* failed to reproduce the binding affinity of the TRIP8b$_{core}$ for the CNBD. This 37 aa long fragment is lacking one important contact residue, namely E$_{240}$, which, in our model, forms a salt bridge with R$_{650}$ of the CNBD. The loss of one crucial interaction is presumably the reason for the major decrease in affinity (about 20 times lower) reported for this peptide.

## TRIP8b$_{nano}$ as a tool for modulating native $I_f$ currents

In functional assays, we showed that TRIP8b$_{nano}$ binds the HCN channel CNBD with high affinity and fully abolishes the cAMP effect in all tested isoforms (HCN1, 2 and 4).

Given the small size of the peptide (<5 kDa), TRIP8b$_{nano}$ is a good candidate for in vivo delivery into intact cells. As a proof of concept, we fused TRIP8b$_{nano}$ to an internalization sequence, the TAT peptide (YGRKKRRQRRRGG). This arginine-rich Cell Penetrating Peptide (CPP) from HIV has been used in several studies as a vehicle for the delivery of large molecules across the plasma membrane (*Guidotti et al., 2017*). In our case, the challenge was to construct a TAT-fusion protein, which would be efficiently delivered in the cell without compromising TRIP8b$_{nano}$ function. Indeed, covalent conjugation of a CPP may negatively affect both the function of the cargo, and the cell-penetrating efficacy of the CPP-peptide chimera (*Kristensen et al., 2016*). The design of the construct was greatly supported by the detailed knowledge of the electrostatic interactions with the target protein CNBD, provided by the NMR model structure. This structure suggested that the polycationic TAT sequence would be best linked to the N-terminus of TRIP8b$_{nano}$ to avoid interference with the cationic residues of CNBD, mainly located in the distal region of C-helix, which are crucial for the binding of the peptide. From test experiments with the TAT-TRIP8b$_{nano}$ peptide in SAN myocytes, we can conclude that this strategy was successful in that: (i) the peptide is efficiently delivered inside the cells; (ii) it is kept in its active conformation; (iii) the TAT sequence did not damage cell membranes and did not interfere with the basic features of $I_f$ and $I_{Ca,L}$ currents; (iv) the modification did not affect the proteolytic stability of the TRIP8b$_{nano}$ peptide at least in the time frame of our experiments (30 min to 1 hr).

In conclusion, we successfully used the miniaturized TRIP8b$_{nano}$ peptide to selectively control native I$_f$ currents and the rate of spontaneous firing in SAN myocytes. Unlike channel blockers, which inhibit ionic currents, the peptide only interferes with the cAMP-based regulation of HCN channels, while leaving basal HCN functions unaltered. In addition and in contrast to even the most selective blockers, it is selective for HCN and it does not interfere with other cAMP-modulated channels present in the SAN, such as L-type Ca$^{2+}$ channels. Collectively, this makes TRIP8b$_{nano}$ a promising tool in targeted therapeutic interventions.

# Materials and methods

## Key resources table

| Reagent type species | Designation | Source or reference | Identifiers | Additional information |
|---|---|---|---|---|
| Gene (human) | HCN1 | Xention Ltd. (Cambridge, UK) | NM_021072.3 | |
| Gene (mouse) | HCN2 | PMID: 11331358 | NM_008226.2 | Laboratory of Steven A. Siegelbaum |
| Gene (rabbit) | HCN4 | PMID: 10212270 | NM_001082707.1 | |
| Gene (mouse) | TRIP8b | PMID: 19555649 | | Laboratory of Steven A. Siegelbaum |
| Strain, strain background (E. coli) | DH5α | Thermo Fisher Scientific | | |
| Strain, strain background (E. coli) | Stbl2 | Thermo Fisher Scientific | | |
| Strain, strain background (Mus musculus) | Male or female C57BL/6J mice | The Jackson Laboratory | RRID:MGI:5653012 | |
| Strain, strain background (Oryctolagus cuniculus) | New Zealand white female rabbits | Envigo | ID strain:HsdOkd:NZW | |
| Cell line (human) | HEK 293T | ATCC | RRID:CVCL_0063 | Tested negative for mycoplasma |
| Biological sample (Mus musculus) | Isolated adult Sinoatrial node (SAN) cardiomyocytes | PMID: 11557233 | | |
| Biological sample (Oryctolagus cuniculus) | Isolated adult Sinoatrial node (SAN) cardiomyocytes | PMID: 2432247 | | |
| Recombinant DNA reagent | pET-52b (plasmid) | EMD Millipore | | |
| Recombinant DNA reagent | modified pET-24b (plasmid) | Laboratory of Daniel L. Minor, Jr. | | |
| Recombinant DNA reagent | pcDNA 3.1 (plasmid) | Clontech Laboratories | | |
| Recombinant DNA reagent | pCI (plasmid) | Promega | | |
| Recombinant DNA reagent | TRIP8bnano (cDNA) | This paper | | Made by PCR and cloning; see Constructs |
| Recombinant DNA reagent | TRIP8bnano (E240R) (cDNA) | This paper | | Made by site-directed mutagenesis of TRIP8bnano wt; see Constructs |
| Recombinant DNA reagent | TRIP8bnano (E241R) (cDNA) | This paper | | Made by site-directed mutagenesis of TRIP8bnano wt; see Constructs |
| Recombinant DNA reagent | TRIP8bnano (E264K) (cDNA) | This paper | | Made by site-directed mutagenesis of TRIP8bnano wt; see Constructs |

*Continued on next page*

*Continued*

| Reagent type species | Designation | Source or reference | Identifiers | Additional information |
|---|---|---|---|---|
| Recombinant DNA reagent | TRIP8bnano (E265K) (cDNA) | This paper | | Made by site-directed mutagenesis of TRIP8bnano wt; see Constructs |
| Recombinant DNA reagent | TRIP8bnano (D252N) (cDNA) | This paper | | Made by site-directed mutagenesis of TRIP8bnano wt; see Constructs |
| Recombinant DNA reagent | TRIP8bcore (cDNA) | PMID: 25197093 | | |
| Recombinant DNA reagent | human HCN2 CNBD (cDNA) | PMID: 25197093 | | |
| Recombinant DNA reagent | human HCN2 CNBD (N547D) (cDNA) | This paper | | Made by site-directed mutagenesis of human HCN2 CNBD wt; see Constructs |
| Recombinant DNA reagent | human HCN2 CNBD (K665E) (cDNA) | This paper | | Made by site-directed mutagenesis of human HCN2 CNBD wt; see Constructs |
| Recombinant DNA reagent | human HCN2 CNBD (K666E) (cDNA) | This paper | | Made by site-directed mutagenesis of human HCN2 CNBD wt; see Constructs |
| Recombinant DNA reagent | human HCN2 CNBD (R650E) (cDNA) | This paper | | Made by site-directed mutagenesis of human HCN2 CNBD wt; see Constructs |
| Recombinant DNA reagent | human HCN1 (cDNA) | Xention Ltd. (Cambridge, UK) | | |
| Recombinant DNA reagent | TRIP8b (1a4) (cDNA) | This paper | | Made by PCR and cloning; see Constructs |
| Recombinant DNA reagent | mouse HCN2 (cDNA) | Laboratory of Steven A. Siegelbaum | | |
| Recombinant DNA reagent | rabbit HCN4 (cDNA) | PMID: 10212270 | | |
| Recombinant DNA reagent | mouse HCN2 (N520D) (cDNA) | This paper | | Made by site-directed mutagenesis of mouse HCN2 wt; see Constructs |
| Sequence-based reagent | human HCN1 (R549E) (cDNA) | This paper | | Made by site-directed mutagenesis of human HCN1 wt; see Constructs |
| Peptide, recombinant protein | TAT-TRIP8b$_{nano}$ (YGRKKRRQRRRGG-NHSLEEEFERAKAAVESTEFWDKMQAEWEEMARRNWISEN) | CASLO ApS | | |
| Peptide, recombinant protein | TAT-(SCRAMBLED)TRIP8b$_{nano}$ (YGRKKRRQRRRGG-RNEAEAAEVAQKDMINERARTHEFEWESWEMWENLSESFK) | CASLO ApS | | |
| Commercial assay or kit | QuickChange Lightning Site-Directed Mutagenesis Kit | Agilent | | |
| Commercial assay or kit | Thermo Scientific TurboFect Transfection Reagent | Thermo Fisher Scientific | | |
| Chemical compound, drug | Adenosine 3', 5'-cyclic monophosphate (cAMP) | SIGMA | | |
| Software, algorithm | Clampfit 10.5/10.7 | Molecular Devices | RRID:SCR_011323 | |
| Software, algorithm | CYANA-2.1 | L. A. Systems, Inc. | | |
| Software, algorithm | AMBER 12.0 | http://pyenmr.cerm.unifi.it/access/index/amps-nmr | | |
| Software, algorithm | HADDOCK2.2 | www.wenmr.eu | | |

## Constructs

The cDNA fragment encoding residues 235–275 (TRIP8b$_{nano}$) of mouse TRIP8b (splice variant 1a4) was cloned into pET-52b (EMD Millipore) downstream of a Strep (II) tag sequence, while the cDNA fragment encoding residues 521–672 of human HCN2 (HCN2 CNBD) was cloned, in a previous study, into a modified pET-24b downstream of a double His$_6$-maltose-binding protein (MBP) (*Saponaro et al., 2014*). The cDNA encoding full-length human HCN1 channel and mouse TRIP8b (1a4) were cloned into the mammalian expression vector pcDNA 3.1 (Clontech Laboratories), while mouse HCN2 channel and rabbit HCN4 channel were cloned into the mammalian expression vector pCI (Promega). Mutations were generated by site-directed mutagenesis (QuikChange site-directed mutagenesis kit; Agilent Technologies) and confirmed by sequencing.

## Preparation of proteins

The HCN2 CNBD WT and mutant proteins, as well as the TRIP8b$_{core}$ and TRIP8b$_{nano}$ proteins (WT and mutants) were produced and purified following the procedure previously described (*Saponaro et al., 2014*).

## Structure calculation of the cAMP-free human HCN2 CNBD in complex with TRIP8b$_{nano}$ and vice versa

NMR experiments were acquired on Bruker Avance III 950, 700 and 500 MHz NMR spectrometers equipped with a TXI-cryoprobe at 298 K. The acquired triple resonance NMR experiments for the assignment of backbone resonances of cAMP-free HCN2 CNBD (CNBD hereafter) in complex with TRIP8b$_{nano}$ and vice versa are summarized in Figure 3-source data 1. $^{15}$N, $^{13}$C′, $^{13}$C$_\alpha$, $^{13}$C$_\beta$, and H$_\alpha$ chemical shifts were used to derive $\phi$ and $\psi$ dihedral angles by TALOS + program (*Cornilescu et al., 1999*) for both CNBD and TRIP8b$_{nano}$. For TRIP8b$_{nano}$, CYANA-2.1 structure calculation (*Güntert and Buchner, 2015*) was performed using 68 $\phi$ and $\psi$ dihedral angles and 40 backbone hydrogen bonds as input. For CNBD, CYANA-2.1 structure calculation was performed using 108 $\phi$ and $\psi$ dihedral angles, combined with the NOEs obtained in our previous determination of the cAMP-free form of the CNBD (*Saponaro et al., 2014*) for those regions not affected by the interaction with TRIP8b$_{nano}$. The 10 conformers of TRIP8b$_{nano}$ and CNBD with the lowest residual target function values were subjected to restrained energy minimization with AMBER 12.0 (*Case, 2012*) (http://pyenmr.cerm.unifi.it/access/index/amps-nmr) and used as input in docking calculations.

## Docking calculations

Docking calculations were performed with HADDOCK2.2 implemented in the WeNMR/West-Life GRID-enabled web portal (www.wenmr.eu). The docking calculations are driven by ambiguous interaction restraints (AIRs) between all residues involved in the intermolecular interactions (*Dominguez et al., 2003*). Active residues of the CNBD were defined as the surface exposed residues (at least 50% of solvent accessibility), which show chemical shift perturbation upon TRIP8b$_{nano}$ binding.

The assignment of the CNBD bound to TRIP8b$_{nano}$ allowed to highlight the residues of CNBD whose backbone featured appreciable Combined Chemical Shift Perturbation (CSP) (*Figure 3—figure supplement 1*). The combined CSP ($\Delta_{HN}$) is given by the equation $\Delta_{HN}=\{((H_{Nfree}-H_{Nbound})^2+((N_{free}-N_{bound})/5)^2)/2\}^{1/2}$ (*Garrett et al., 1997*).

Passive residues of CNBD were defined as the residues close in space to active residues and with at least 50% solvent accessibility.

In the case of TRIP8b$_{nano}$, the conserved stretch E$_{239}$-E$_{243}$, located in helix N, was defined as active region in a first docking calculation, while all the other solvent accessible residues of the peptide were defined as passive. This docking calculation generated several clusters. A post-docking filter step allowed us to select those clusters having an orientation of TRIP8b$_{nano}$ bound to CNBD in agreement with a DEER study on the CNBD - TRIP8b$_{nano}$ interaction (*DeBerg et al., 2015*). The selected clusters grouped in two classes on the basis of the orientation of helix N of TRIP8b$_{nano}$ (N) relative to CNBD (*Figure 3—figure supplement 2*. A second docking calculation was subsequently performed introducing also residues E$_{264}$-E$_{265}$, located in helix C of TRIP8b$_{nano}$ as active residues. The active residues for CNBD were the same used for the first calculation. For this second HADDOCK calculation, 14 clusters were obtained and ranked according to their HADDOCK score.

Among them only four clusters showed both an orientation of TRIP8b$_{nano}$ bound to CNBD in agreement with the DEER study (*DeBerg et al., 2015*) and the involvement of $E_{239}$-$E_{243}$ stretch of TRIP8b$_{nano}$ in the binding to CNBD. These clusters were manually analyzed and subjected to a per-cluster re-analysis following the protocol reported in http://www.bonvinlab.org/software/haddock2.2/analysis/#reanal. From this analysis, it resulted that the top-ranking cluster, i.e. the one with the best energetic and scoring functions, has a conformation in agreement with mutagenesis experiments (*Figure 3—figure supplement 3*). Energy parameters (van der Waals energy, electrostatic energy, desolvation energy, and the penalty energy due to violation of restraints) for this complex model are reported in *Figure 3—source data 2*.

Both docking calculations were performed using 10 NMR conformers of both the CNBD and the TRIP8b$_{nano}$ structures calculated as described above. In the TRIP8b$_{nano}$ structures the unfolded N- and C-terminal regions were removed, while in the CNBD structures only the unfolded N-terminal region was removed. This is because the C-terminal region of the CNBD is known to comprise residues involved in TRIP8b$_{nano}$ binding (*Saponaro et al., 2014*). Flexible regions of the proteins were defined based on the active and passive residues plus two preceding and following residues. The residue solvent accessibility was calculated with the program NACCESS (*Hubbard and Hornton, 1993*). In the initial rigid body docking calculation phase, 5000 structures of the complex were generated, and the best 400 in terms of total intermolecular energy were further submitted to the semi-flexible simulated annealing and a final refinement in water. Random removal of the restraints was turned off. The number of flexible refinement steps was increased from the default value of 500/500/1000/1000 to 2000/2000/2000/4000. The final 400 structures were then clustered using a cutoff of 5.0 Å of RMSD to take into consideration the smaller size of protein-peptide interface.

## Electrophysiology of HEK 293 T cells

HEK 293 T cells were cultured in Dulbecco's modified Eagle's medium (Euroclone) supplemented with 10% fetal bovine serum (Euroclone), 1% Pen Strep (100 U/mL of penicillin and 100 µg/ml of streptomycin), and stored in a 37°C humidified incubator with 5% $CO_2$. The plasmid containing cDNA of wild-type and mutant HCN1, HCN2 and HCN4 channels (1 µg) was co-transfected for transient expression into HEK 293 T cells with a plasmid containing cDNA of Green Fluorescent Protein (GFP) (1.3 µg). For co-expression with TRIP8b (1a-4), HEK 293 T cells were transiently transfected with wild-type (wt) and/or mutant human HCN1 cDNA (1 µg), wt TRIP8b (1a-4) cDNA (1 µg) and cDNA of Green Fluorescent Protein (GFP) (0.3 µg).

One day after transfection, GFP-expressing cells were selected for patch-clamp experiments in whole-cell configuration. The experiments were conducted at R.T. The pipette solution in whole cell experiments contained: 10 mM NaCl, 130 mM KCl, 1 mM egtazic acid (EGTA), 0.5 mM $MgCl_2$, 2 mM ATP (Mg salt) and 5 mM HEPES–KOH buffer (pH 7.4). The extracellular bath solution contained 110 mM NaCl, 30 mM KCl, 1.8 mM $CaCl_2$, 0.5 mM $MgCl_2$ and 5 mM HEPES–KOH buffer (pH 7.4).

TRIP8b$_{nano}$ was added (10 µM) to the pipette solution. cAMP was added at different concentration to the pipette solution depending on the HCN isoform used: 0 µM for HCN1, 5 µM for HCN2 and 1 µM for HCN4.

Whole-cell measurements of HCN channels were performed using the following voltage clamp protocol depending on the HCN isoform measured: for HCN1, holding potential was –30 mV (1 s), with steps from –20 mV to –120 mV (10 mV interval, 3.5 s) and tail currents recorded at –40 mV (3 s); for HCN2, holding potential was –30 mV (1 s), with steps from –40 mV to –130 mV (10 mV interval, 5 s) and tail currents recorded at −40 mV (5 s); for HCN4, holding potential was –30 mV (1 s), steps from –30 mV to –165 mV (15 mV interval, 4.5 s) and tail currents were recorded at −40 mV (5 s). Current voltage relations and activation curves were obtained by the above activation and deactivation protocols and analyzed by the Boltzmann equation, see data analysis.

## Isolation and electrophysiology of rabbit sinoatrial node cells

Animal protocols conformed to the guidelines of the care and use of laboratory animals established by Italian and European Directives (D. Lgs n° 2014/26, 2010/63/UE). New Zealand white female rabbits (0.8–1.2 kg) were anesthetized (xylazine 5 mg/Kg, i.m.), and euthanized with an overdose of sodium thiopental (i.v.); hearts were quickly removed, and the SAN region was isolated and cut in small pieces. Single SAN cardiomyocytes were isolated following an enzymatic and mechanical

procedure as previously described (*DiFrancesco et al., 1986*). Following isolation, cells were maintained at 4°C in Tyrode solution: 140 mM NaCl, 5.4 mM KCl, 1.8 mM CaCl$_2$, 1 mM MgCl$_2$, 5.5 mM D-glucose, 5 mM HEPES-NaOH (pH 7.4).

For patch clamp experiments cells were placed in a chamber on an inverted microscope and experiments were performed in the whole-cell configuration at 35 ± 0.5°C. The pipette solution contained: 10 mM NaCl, 130 mM KCl, 1 mM egtazic acid (EGTA), 0.5 mM MgCl$_2$, and 5 mM HEPES–KOH buffer (pH 7.2). The I$_f$ current was recorded from single cells superfused with Tyrode solution with 1 mM BaCl$_2$, and 2 mM MnCl$_2$.

I$_f$ activation curves were obtained using a two-step protocol in which test voltage steps (from −30 to −120 mV, 15 mV interval) were applied from a holding potential of −30 mV and were followed by a step to −125 mV. Test steps had variable durations so as to reach steady –state activation at all voltages. Analysis was performed with the Boltzmann equation (see data analysis).

In current-clamp studies, spontaneous action potentials were recorded from single cells superfused with Tyrode solution, and rate was measured from the interval between successive action potential. When indicated cAMP (1 μM) and/or nanoTRIP8b (10 μM) were added to the pipette solution.

## Isolation and electrophysiology of mouse sinoatrial node cells

Mice were killed by cervical dislocation under general anesthesia consisting of 0.01 mg/g xylazine (2% Rompun; Bayer AG), 0.1 mg/g ketamine (Imalgène; Merial) and 0.04 mg/g of Na-pentobarbital (Euthanasol VET, Laboratoire TVM, Lempdes, France), and beating hearts were quickly removed. The SAN region was excised in warmed (35°C) Tyrode's solution containing: 140 mM NaCl, 5.4 mM KCl, 1.8 mM CaCl$_2$, 1 mM MgCl$_2$, 1 mM Hepes-NaOH (pH = 7.4), and 5.5 mM D-glucose and cut in strips. Strips were then transferred into a 'low-Ca$^{2+}$-low-Mg$^{2+}$' solution containing: 140 mM NaCl; 5.4 mM KCl, 0.5 mM MgCl$_2$, 0.2 mM CaCl$_2$, 1.2 mM KH$_2$PO$_4$, 50 mM taurine, 5.5 mM D-glucose, 1 mg/ml bovine serum albumin (BSA), 5 mM Hepes-NaOH (pH = 6.9).

Tissue was digested by adding Liberase TH (0.15 mg/ml, Roche Diagnostics GmbH, Mannheim, Germany), elastase (1.9 U/ml, Worthington, Lakewood). Digestion was carried out for a variable time of 15–18 min at 35°C. Tissue strips were then washed and transferred into a modified 'Kraftbrühe' (KB) medium containing: 70 mM L-glutamic acid, 20 mM KCl, 80 mM KOH, (±) 10 mM D- b-OH-butyric acid; 10 mM KH$_2$PO$_4$, 10 mM taurine, 1 mg/ml BSA and 10 mM Hepes-KOH (pH = 7.4).

Single SAN cells were isolated by manual agitation in KB solution at 35°C for 30–50 s.

Cellular automaticity was recovered by re-adapting the cells to a physiological extracellular Ca$^{2+}$ concentration by addition of a solution containing: 10 mM NaCl, 1.8 mM CaCl$_2$ and normal Tyrode solution containing BSA (1 mg/ml). The final storage solution contained: 100 mM NaCl, 35 mM KCl, 1.3 mM CaCl$_2$, 0.7 mM MgCl$_2$, 14 mM L-glutamic acid, (±) 2 mM D-b-OH-butyric acid, 2 mM KH$_2$PO$_4$, 2 mM taurine, 1 mg/ml BSA, (pH = 7.4). Cells were then stored at room temperature until use. All chemicals were from SIGMA (St Quentin Fallavier, France).

For electrophysiological recording, SAN cells in the storage solution were harvested in special custom-made recording plexiglas chambers with glass bottoms for proper cell attachment and mounted on the stage of an inverted microscope (Olympus IX71) and perfused with normal Tyrode solution. The recording temperature was 36°C. We used the whole-cell variation of the patch-clamp technique to record cellular ionic currents, by employing a Multiclamp 700B (Axon Instruments Inc., Foster USA) patch clamp amplifier. Recording electrodes were fabricated from borosilicate glass, by employing a WZ DMZ-Universal microelectrode puller (Zeitz-Instruments Vertriebs GmbH, Martinsried, Germany).

I$_f$ was recorded under standard whole-cell configuration during perfusion of standard Tyrode's containing 2 mM BaCl$_2$ to block I$_{K1}$. Patch-clamp pipettes were filled with an intracellular solution containing: 130 mM KCl, 10 mM NaCl, 1 mM EGTA, 0.5 mM MgCl$_2$ and 5 mM HEPES (pH 7.2).

For recording of L-type Ca$^{2+}$ currents, pipette solution contained: 125 mM CsOH, 20 mM tetraethylammonium chloride (TEA-Cl), 1.2 mM CaCl$_2$, 5 mM Mg-ATP, 0.1 mM Li$_2$-GTP, 5 mM EGTA and 10 mM HEPES (pH 7.2 with aspartate). 30 μM TTX (Latoxan, Portes lès Valence, France) to block INa was added to external solution containing: 135 mM tetraethylammonium chloride (TEA-Cl), 4 mM CaCl$_2$,10 mM 4-amino-pyridine, 1 mM MgCl$_2$, 10 mM HEPES and 1 mg/ml Glucose (pH 7.4 with TEA-OH).

Electrodes had a resistance of about 3 MΩ. Seal resistances were in the range of 2–5 GΩ. 10 µM TRIPb8$_{nano}$ was added to pipette solution. 10 µM TAT-TRIPb8$_{nano}$ was added in cell storage solution for at least 30 min before patch clamp recording.

## TAT-peptides

TAT-TRIP8b$_{nano}$ (**YGRKKRRQRRRGG**-NHSLEEEFERAKAAVESTEFWDKMQAEWEEMARRNWISEN, TAT sequence is shown in bold type) and TAT-(SCRAMBLED)TRIP8b$_{nano}$ (**YGRKKRRQRRRGG**-RNEAEAAEVAQKDMINERARTHEFEWESWEMWENLSESFK, TAT sequence is shown in bold type) were purchased from CASLO ApS. TAT-peptides were dissolved in Milliq water (1.5 mM) and added to the petri dish at the final concentration (10 µM) 30 min before current recordings. During the patch clamp experiments, cells were perfused with standard Tyrode with 2 mM BaCl$_2$ (see above) without the peptides. Recordings from the same petri dish were performed over a time window of 10 to 60 min in peptide-free solution

## Data analysis

Data were acquired at 1 kHz using an Axopatch 200B amplifier and pClamp10.5 or 10.7 software (Axon Instruments). Data were analyzed off-line using Clampfit 10.5 or 10.7 (Molecular Devices) and Origin 2015 or 16 (OriginLab Corp., Northampton MA). Activation curves were analyzed by the Boltzmann equation, $y = 1/\{1 + \exp[(V-V_{1/2})/s]\}$, where y is fractional activation, V is voltage, $V_{1/2}$ half-activation voltage, and s the inverse slope factor (mV) (*DiFrancesco, 1999*). Mean activation curves were obtained by fitting individual curves from each cell to the Boltzmann equation and then averaging all curves obtained. Activation time constants ($\tau_{on}$) were obtained by fitting a single exponential function,

$I = I_0 \exp(-t/\tau)$ to current traces recorded at the indicated voltages.

## Ethics statement

Experiments on rabbit SAN cells were performed using left-over cells obtained during experiments approved by the Animal Welfare Body of the University of Milan and by the Italian Ministry of Health (license n.1127/2015-PR). Animal procedures were conformed to the guidelines of the care and use of laboratory animals established by Italian and European Directives (D. Lgs n° 2014/26, 2010/63/UE).

Mouse primary pacemaker cells were isolated from adult C57BL/6J mice as previously described (Mangoni and Nargeot, Cardiovasc Res 2001), in accordance with the Guide for the Care and Use of Laboratory Animals (eighth edition, 2011), published by the US National Institute of Health and European directives (2010/63/EU). The protocol was approved by the ethical committee of the University of Montpellier and the French Ministry of Agriculture (protocol N°: 2017010310594939).

## Acknowledgements

This work has been supported by Fondazione CARIPLO grant 2014–0796 to AM, BS and LB, by 2016 Schaefer Research Scholars Program of Columbia University to AM, by European Research Council (ERC) 2015 Advanced Grant 495 (AdG) n. 695078 noMAGIC to AM and GT, by National Institutes for Health Grant R01 NS036658 to BS, by Instruct-ERIC and national member subscriptions to LB, and by Accademia Nazionale dei Lincei (Giuseppe Levi foundation) to AS. We specially thank the EU ESFRI Instruct Core Centre CERM-Italy.

## Additional information

### Funding

| Funder | Grant reference number | Author |
| --- | --- | --- |
| Accademia Nazionale dei Lincei | Postdoctoral Fellowship in Neurobiology research | Andrea Saponaro |
| H2020 European Research Council | 2015 Advanced Grant 495 n. 695078 noMAGIC | Gerhard Thiel Anna Moroni |

| Fondazione Cariplo | 2014-0796 | Lucia Banci<br>Bina Santoro<br>Anna Moroni |
|---|---|---|
| Instruct-ERIC | | Lucia Banci |
| National Institutes of Health | R01 NS036658 | Bina Santoro |
| Columbia University | 2016 Schaefer Research<br>Scholars Program | Anna Moroni |

The funders had no role in study design, data collection and interpretation, or the decision to submit the work for publication.

## Author contributions

Andrea Saponaro, Conceptualization, Investigation, Methodology, Writing—original draft; Francesca Cantini, Formal analysis, Investigation, Methodology; Alessandro Porro, Annalisa Bucchi, Vincenzo Maione, Formal analysis, Investigation; Dario DiFrancesco, Methodology; Chiara Donadoni, Bianca Introini, Investigation; Pietro Mesirca, Data curation, Formal analysis, Investigation; Matteo E Mangoni, Conceptualization, Data curation, Writing—original draft; Gerhard Thiel, Conceptualization, Supervision, Writing—original draft; Lucia Banci, Conceptualization, Data curation, Supervision; Bina Santoro, Conceptualization, Supervision, Funding acquisition, Writing-original draft; Anna Moroni, Conceptualization, Resources, Supervision, Funding acquisition, Methodology, Writing—original draft

## Author ORCIDs

Andrea Saponaro  http://orcid.org/0000-0001-5035-5174
Francesca Cantini  http://orcid.org/0000-0003-0526-6732
Vincenzo Maione  http://orcid.org/0000-0002-8229-6612
Matteo E Mangoni  https://orcid.org/0000-0002-8892-3373
Bina Santoro  http://orcid.org/0000-0002-4277-1992
Anna Moroni  http://orcid.org/0000-0002-1860-406X

## Ethics

Animal experimentation: Experiments on rabbit SAN cells were performed using left-over cells obtained during experiments approved by the Animal Welfare Body of the University of Milan and by the Italian Ministry of Health (license n.1127/2015-PR). Animal procedures were conformed to the guidelines of the care and use of laboratory animals established by Italian and European Directives (D. Lgs no 2014/26, 2010/63/UE). Mouse primary pacemaker cells were isolated from adult C57BL/6J mice as previously described (Mangoni and Nargeot, Cardiovasc Res 2001), in accordance with the Guide for the Care and Use of Laboratory Animals (eighth edition, 2011), published by the US National Institute of Health and European directives (2010/63/EU). The protocol was approved by the ethical committee of the University of Montpellier and the French Ministry of Agriculture (protocol N°: 2017010310594939).

## Decision letter and Author response

Decision letter https://doi.org/10.7554/eLife.35753.021
Author response https://doi.org/10.7554/eLife.35753.022

# Additional files

## Supplementary files

• Transparent reporting form
DOI: https://doi.org/10.7554/eLife.35753.019

## Data availability

All data generated or analysed during this study are included in the manuscript and supporting files. Source data files have been provided for Figure 3.

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
