## [Decision Letter]

Thank you for submitting your article "Structure-guided design of a synthetic peptide for orthogonal control of HCN channels" for consideration by *eLife*. Your article has been reviewed by three peer reviewers, including Volker Dötsch as the Reviewing Editor and Reviewer #1, and the evaluation has been overseen by Richard Aldrich as the Senior Editor. The following individual involved in review of your submission has agreed to reveal their identity: Catherine Proenza (Reviewer #3).

The reviewers have discussed the reviews with one another and the Reviewing Editor has drafted this decision to help you prepare a revised submission.

Summary:

The manuscript entitled "Structure-guided design of a synthetic peptide for orthogonal control of HCN channels," by Moroni et al. investigates a novel, rationally engineered polypeptide for the control of HCN channels in vivo. The group utilized an endogenous regulator of HCN channels (TRIP8b) to design their 40 amino acid functional regulator (TRIP8b_nano_). This peptide antagonizes the binding of cAMP to the HCN channels. The group proceeds to use isothermal calorimetry, NMR spectroscopy and reverse charge mutations to define the interaction sites and structure between HCN channels and the TRIP8b_nano_. Finally, the group uses the TRIP8b_nano_ to modulate HCN channels and their resulting currents in both HEK 293 cells and in rabbit and mouse SAN cells. Of particular interest therapeutically, the TAT internalization sequence was used for application of the TRIP8b_nano_ to a portion of the rabbit SAN cells. This work is important as specific modulation of I_f_ has potential utility therapeutically, especially when compared to current ion channel pore blockers.

Essential revisions:

1) What is the justification for utilizing reverse charge mutations as opposed to alanine mutation? It seems like a reverse charge mutation biases the system toward less interaction between the HCN and TRIP8b_nano_.

2) Is there any data on how long the TRIP8b_nano_ is able to maintain an effect in vivo?

3) In the Discussion it is stated that the TRIP8b_nano_ is entirely specific for HCN channels. Is this too strong of an assertion? Is there any data regarding off target effects on non-ion channel targets?

4) Include legends in some of the figures for clarification. (Figure 4B, Figure 5B).

5) The peptide experiments should be supported by controls using scrambled peptides. This is particularly important for charged TAT peptides, which can have non-specific effects.

6) Related to this, what is the evidence that the TAT-fused peptide indeed translocates into the cell? Have control experiments with the non-TAT-fused peptide been performed? These should not be able to penetrate the cell.

---

## [Author Response]

Essential revisions:1) What is the justification for utilizing reverse charge mutations as opposed to alanine mutation? It seems like a reverse charge mutation biases the system toward less interaction between the HCN and TRIP8b_nano_.

As shown in Figure 3, the CNBD ‐ TRIP8b_nano_ interaction is mainly driven by an electrostatic network rather than single, isolated salt bridges. Therefore, our choice to invert charges instead of neutralize them was to overcome possible compensatory effects deriving from the network of electrostatic interactions. The reviewer is certainly right, a reverse charge mutation biases the system toward less interaction. But in the case of three negative charges (D_257_, E_264_, E_265_) facing three positive ones (R_662_, K_665_, K_666_), we considered the possibility of compensatory effects form the neighbors very likely. Similar consideration holds also for E_241_ and E_240_ although we could have performed R_650_A. In this case we decided not to, for coherence with the rest.

2) Is there any data on how long the TRIP8b_nano_ is able to maintain an effect in vivo?

Cells were pretreated by adding the cell‐penetrating peptides TAT‐TRIP8b_nano_ and TAT‐ (SCRAMBLED) TRIP8b_nano_ (see below our reply to point 5) but also TRIP8b_nano_ (see our reply to point 6) to the extracellular solution, 30 minutes before the experiment. At the end of the pretreatment, cells were patched under constant perfusion of extracellular solution that did not contain the (TAT)‐peptides. Our patch recordings nevertheless show that TAT‐TRIP8b_nano_ was still effective even after 60 minutes from the beginning of the experiment. This indicates in our view that TAT‐TRIP8b_nano_, once accumulated inside, is not released from the cell and maintains its effect up to 1 hours without becoming toxic.

We have added an explanatory paragraph on the modality of the experiment with TAT‐peptides (Materials and methods, subsection “TAT-peptides”).

3) In the Discussion it is stated that the TRIP8b_nano_ is entirely specific for HCN channels. Is this too strong of an assertion? Is there any data regarding off target effects on non-ion channel targets?

In agreement with the reviewer we have rephrased the sentence as follows:

“…it is selective for HCN and it does not interfere with other cAMP‐modulated channels present in the SAN, such as L‐type Ca^2+^channels. Collectively, this makes TRIP8b_nano_ a promising tool in targeted therapeutic interventions”.

4) Include legends in some of the figures for clarification. (Figure 4B, Figure 5B).

Not clear what the request was. We interpret it as the need to add, in the legends, the cross‐reference to data analysis in the Materials and methods (i.e. Boltzmann fit). Hope we got it right.

5) The peptide experiments should be supported by controls using scrambled peptides. This is particularly important for charged tat peptides, which can have non-specific effects.

We have tested the scrambled peptide on mouse SAN cells. The results are included as new Figure 6—figure supplement 1.

6) Related to this, what is the evidence that the TAT-fused peptide indeed translocates into the cell? Have control experiments with the non-TAT-fused peptide been performed? These should not be able to penetrate the cell.

We further tested the TRIP8b_nano_ peptide in HEK 293T cells. Also in this case, the peptide added to the extracellular solution for 30 minutes, did not affect the current. We added this result to Figure 4D.